# Clinical Characteristics of Ciguatera Poisoning in Martinique, French West Indies—A Case Series

**DOI:** 10.3390/toxins14080535

**Published:** 2022-08-03

**Authors:** Dabor Résière, Jonathan Florentin, Hossein Mehdaoui, Zakaria Mahi, Papa Gueye, Didier Hommel, Jean Pujo, Flaubert NKontcho, Patrick Portecop, Rémi Nevière, Hatem Kallel, Bruno Mégarbane

**Affiliations:** 1Intensive Care Unit, Martinique University Hospital, 97261 Fort-de-France, Martinique, France; jonathan.florentin@chu-martinique.fr (J.F.); hossein.mehdaoui@chu-martinique.fr (H.M.); papa.gueye@chu-martinique.fr (P.G.); 2Intensive Care Unit, Cayenne General Hospital, 97300 Cayenne, French Guiana, France; didier.hommel@ch-cayenne.fr (D.H.); hatem.kallel@ch-cayenne.fr (H.K.); 3Emergency Department, Cayenne General Hospital, 97300 Cayenne, French Guiana, France; mahi.zakaria@gmail.com (Z.M.); jean.pujo@ch-cayenne.fr (J.P.); 4Pharmacy Department, Cayenne General Hospital, 97300 Cayenne, French Guiana, France; flaubert.nkontcho@ch-cayenne.fr; 5Emergency Department, University Hospital of Guadeloupe, 97100 Pointe-à-Pitre, Guadeloupe, France; patrick.portecop@chu-guadeloupe.fr; 6Department of Cardiology, University Hospitals of Martinique, Pierre Zobda-Quitman Hospital, 97261 Fort-de-France, Martinique, France; remi.neviere@chu-martinique.fr; 7Department of Medical and Toxicological Critical Care, Lariboisière Hospital, Paris Cité University, INSERM UMRS1144, 75006 Paris, France

**Keywords:** ciguatera, ciguatoxin, fish, French West Indies, Martinique, poisoning

## Abstract

Ciguatera poisoning (CP) is one of the most common causes worldwide of marine poisoning associated with fish consumption from tropical areas. Its incidence is underreported. CP cases seem to increase with grouped cases reported during summer. Exposure to ciguatoxins, toxins responsible for CP with sodium-channel agonistic, voltage-gated potassium channel blocking, cholinergic, and adrenergic activities, may result in a large spectrum of manifestations. We aimed to describe the clinical characteristics, management, and outcome of CP in Martinique, French West Indies. We conducted an observational retrospective single-center study during six years (October 2012 to September 2018) including all CP patients managed by the prehospital medical services, admitted to the university hospital emergency department, or declared to the regional health agency. A total of 149 CP patients (81 females/63 males; median age, 46 years (interquartile range, 34–61)) were included. Acute features consisted in general (91%; mainly, myalgia pruritus, and asthenia), gastrointestinal (90%; mainly diarrhea, abdominal pain, and nausea), neurological (72%; mainly, paresthesia, dysgeusia, and impairment of hot/cold feeling), and cardiovascular manifestations (22%; bradycardia, hypotension, and heart conduction disorders). Management was supportive. No patient died but symptoms persisted in 40% of the 77 patients with follow-up at day 15. CP was mainly attributed to the ingestion of trevallies (59%), snappers (13%), and king mackerels (8%) with collective contaminations (71%). Unusual fish (tuna, salmon, and spider conchs) were suspected in rare cases. Ingestion of trevallies was associated with significantly higher persistent symptoms (odds ratio, 3.00; 95% confidence interval, (1.20–8.00); *p* = 0.03). CP incidence was 0.67 cases per 10,000 patient-years in Martinique over the study period. To conclude, CP represents an increasing public health issue in Martinique, as is the case in other Caribbean islands. Patients present usual but possibly life-threatening features. Outcome is excellent despite frequently prolonged manifestations.

## 1. Introduction

Ciguatera poisoning (CP) is one of the most common marine poisonings related to fish consumption worldwide [1,2,3]. CP mostly results from accidental ingestion of fish flesh contaminated with the toxins produced by two dinoflagellate genera called *Gambierdiscus* spp. and *Fukuoya* spp. [4]. These unicellular algae may grow on the surface of macroalgae, whose proliferation is usually limited in the area where live corals are developed. However, if coral is destroyed for artificial or natural reasons, as more and more frequently observed with cyclones, sand fog, volcanic activity, sargassum waves, pollution, and work on the reefs [3,4,5,6], certain fast-growing algae may increase, stimulating the development of *Gambierdiscus* spp. and *Fukuoya* spp. These algae secrete several toxins including ciguatoxins, colorless and odorless heat-stable non-protein toxins, which find their way into the food chain. Among fish species likely to transmit ciguatoxins, predators such as jacks, groupers, trevallies, grazers, and barracudas are the most frequently involved.

Incidence of CP has been determined in various intertropical areas [5,6,7,8,9,10] (See Table A1). Although underestimated with approximately 50,000 to 500,000 annual suspected cases reported worldwide, its incidence has sharply increased during the last decades in the West Indies [5]. CP-attributed manifestations are variable and non-specific, mostly beginning within minutes following exposure with gastrointestinal disorders followed by general, neurological, and cardiovascular disorders [11,12,13]. They are related to the sodium-channel agonistic, voltage-gated potassium channel blocking, cholinergic, and adrenergic activities of ciguatoxin [1,2,3]. Management is supportive and outcomes are favorable.

A wide range of symptoms and regional characteristics have been attributed to CP. As one of the reasons, differences in types of ciguatoxins have been suspected [14]. Four types of ciguatoxins have been described according to their carbon skeleton and origin. The Pacific ciguatoxins (P-CTX) from the Pacific Ocean, the Caribbean-ciguatoxin (C-CTX) found in Caribbean Sea and the Indian-ciguatoxin (I-CTX) in the Indian Ocean are the most prevalent types [15]. All these toxins bind to site five of the α-subunit of the voltage gated sodium channels (VGSC) causing a negative shift in the voltage dependence of the channel activation and a concentration–dependent decrease in the maximum inward sodium current, which increases cell excitability [3]. However, due to their variable structures, ciguatoxins could give different modes of action and result in variable clinical expression.

To date, the incidence and presentations of CP in Martinique, the largest French island in the West Indies, are poorly known. Therefore, we designed this study aiming to describe the clinical characteristics (primary objective) and evaluate management and outcome of CP patients (secondary objective).

## 2. Results

Overall, 149 patients were diagnosed with CP in Martinique during the six-year period (from 2012 to 2018) and included in the study, allowing calculation of an incidence of 0.67 cases/10,000 patient-years over the study period in an estimated population of 368,783 inhabitants, according to the French National Institute for Statistics and Economic Studies in January 2018 [16]. These CP patients were 81 females/63 males, with a median age of 46 years (interquartile range, 34–61). About 15% presented a history of hypertension. Overall, 12 patients were managed by the prehospital medical services, 66 patients were admitted to the emergency department, and 117 patients were reported by the Regional Health Agency (RHA). Three patients (2%) had a previous CP history.

CP resulted from collective food poisoning in 106/149 cases (71%), involving a median number of three contaminated guests (three–five) per event with fish mainly derived from local Martinique fisheries (83%). The main involved geographical areas were the counties of Vauclin and Trinité (together accounting for 45% of the cases) (See Figure A1). Nineteen CP cases occurred in 2017 in relation to the consumption of snappers imported from the Indian Ocean area and sold in supermarkets. In August and September 2018, two other episodes of collective food poisoning, attributed to the consumption of fish from local fisheries, involved eight and nine patients, respectively.

CP symptoms occurred 5 h (3–8) following fish ingestion. Manifestations were general (135/149, 91%), gastrointestinal (134/149, 90%), neurological (108/149, 72%), and/or cardiovascular (33/149, 22%) (Table 1). Vital parameters available in 66/149 patients (44%) admitted to the emergency department showed body temperature ≤ 36.5 °C (26/66, 39%), heart rate ≤ 40 bpm (8/66, 12%), and systolic blood pressure ≤80 mmHg (8/66, 12%). Electrocardiogram showed cardiac conduction disturbances (9/66 patients, 14%) including first/second-degree atrioventricular (*n* = 5) and bundle branch blocks (*n* = 4). One additional patient exhibited slow atrial fibrillation. Blood lactate concentration was elevated >2 mmol/L in four patients, suggestive of tissue hypoxia in relation to cardiovascular failure.

In the 66 patients transported to the emergency department, management was supportive (64%) and included intravenous atropine (10/66, 15%) and mannitol (11/66, 17%) (Table 2). In the other 83 not transported patients, no particular treatment was required or prescribed on the scene. Overall, hospitalization was required (24/149, 16%), mainly in the post-emergency hospitalization unit (20/24, 83%). Length of hospital stay was 1 day (1–2). Follow-up at day 15 after consumption was available in 77/149 patients (52%), including 39 hospital-managed patients followed-up in the clinical toxicology consultation and 38 patients followed-up by phone call. Reported persistent symptoms (31/77, 40%) consisted of non-specific aches including myalgia and arthralgia (20/77, 26%), neurological complains (15/77, 20%), asthenia (11/77, 14%), and whole-body pruritus predominantly on the palms of the hands and the soles of the feet (10/77, 13%). No patient died in relation to CP.

The incriminated fish species included trevallies (Carangidae), snappers (Lutjanus buccanella), king mackerels (Scomberomorus regalis), and moray eels (Gymnothorax funebris) (Table 3). Rare cases were attributed to tunas (Thunnus atlanticus). Patients declared to have eaten only the fish flesh (116/149, 78%) but also the head or viscera (25/149, 17%) and the tail (12/149, 8%). Intoxication with trevallies resulted in significantly more frequent acute gastrointestinal manifestations (odds ratio (OR), 3.16; 95% confidence interval (CI) (1.52–6.56); *p* = 0.03) and persistent symptoms (OR, 3.00; 95% CI, (1.20–8.00); *p* = 0.03) (Table 4). Of note, the two patients with previous CP history did not exhibit more severe symptoms and did not consume uncommon fish.

## 3. Discussion

### 3.1. Incidence of CP in the World

The epidemiology of CP is challenging to assess. It has been estimated that less than 20% of CP cases are reported [5]. In the West Indies, CP incidence over the 1996–2006 period was estimated at 0.2/10,000 patient-years in Martinique, 0.3/10,000 patient-years in Guadeloupe, 19.9/10,000 patient-years in the British Virgin Islands 34.4/10,000 patient-years in Antigua, and 58.6/10,000 patient-year in Montserrat [5]. An annual incidence of 1.47/10,000 patient-years (95% CI, (1.29–1.66)) was determined in 2016 in Guadeloupe, i.e 5 times the previously reported incidence in 1996–2006 [7]. In the other intertropical French territories, incidence of CP was estimated at 0.2/10,000 patient-years in 2000–2010 in Reunion (Indian Ocean) [6] and 18/10,000 patient-years in 2016 in French Polynesia (Pacific Ocean), the most affected territory in France [9]. In the US, Florida is the state with the highest incidence due to its intertropical geographical location, with an estimated incidence of 0.56/10,000 patient-years in 2000–2011 [8].

The incidence reported in our study was 3 times higher than that reported during 1996–2006 (0.2/10,000 patient-years) [5]. The estimate for the 1996–2006 period was based only on RHA reports whereas our study collected data from cases observed in the emergency department. If only cases reported to RHA (*n* = 117) were taken into account, the incidence would have been 0.52 cases/10,000 patient-years. The increased incidence of CP in Martinique, which we reported similarly to Guadeloupe, the second main French island in the West Indies [7], could at least partly be related to the improved data collection. However, it may also correspond to an actual increase in incidence, even if still probably underestimated. Interestingly, tourism in intertropical countries is likely to increase CP cases in Europe and North America among travelers who have stayed in endemic areas. For instance, six CP cases involving 20 persons were reported in 2010–2011 in the state of New York, in relation to the consumption of imported fish [17].

Interestingly, collective food poisoning was responsible for 71% of our cases, mainly in summer, compared to 41% in Polynesia [18] and 32% in Guadeloupe over the 2013–2016 period [7]. Grouped cases of CP are increasingly reported such as thirty-four cases in Germany in 2009–2012 [19,20]; thirteen patients in Italy in 1995–1999 [21]; thirty patients following barracuda ingestion, in 1995 and eighteen others in 1997–2002 in France, especially in the Mediterranean city of Marseille [22].

### 3.2. Clinical Characteristics of CP

CP is a polymorphic condition with no specific toxidrome, responsible for ~175 symptoms with marked variable intensity between individuals [23]. The variation in symptom patterns in the Pacific Ocean, Indian Ocean, and Caribbean Sea reflects geographic differences in ciguatoxin profiles and individual risk factors among patients in the different parts of the world [5]. In the Caribbean area, gastrointestinal manifestations are characteristic in the acute phase, followed by neurological, mainly peripheral, manifestations. By contrast, in the Pacific and Indian Ocean regions, neurological manifestations are more prevalent and severe in the acute phase.

CP presentations in our series were similar to presentations described in other Caribbean studies [5,6,7,24]. Gastrointestinal manifestations are foremost, including nausea/vomiting, intense abdominal pain, and profuse liquid diarrhea without bloody glair. Cardiovascular impairments may appear rapidly and become life threatening. They were present in 22% of our patients versus 41% in the Guadeloupe cohort [7] and 12% in the French Guyana series [25]. The relatively high frequency of cardiovascular disorders in our study may be attributed in part to the study design, which mainly included patients admitted to the emergency department, who corresponded to the most severe cases. It is also possible that bradycardia or hypotension were discriminating factors that allowed CP diagnosis despite a possible underestimation due to the lack of knowledge of this condition by emergency physicians.

Neurological manifestations are polymorphous but suggestive of CP [26,27,28]. They usually occur within days following fish ingestion. The main impairments we found were paresthesia (especially perioral), dysesthesia with inversion hot/cold, sensation of metallic taste, and diffuse hyperesthesia. Interestingly, whereas sometimes acknowledged as pathognomonic of CP [27], reversal of hot and cold was only observed in 16% of our patients, underlying the rarity of specific clinical symptoms or signs that may unambiguously ascertain clinical diagnosis. Studies suggested that unusual sensations represent tingling or “electric shock” pain rather than a true reversal of hot and cold perception [29]. Visual disorders and ataxic disorders have also been described, mimicking Guillain-Barre syndrome [26]. Psychiatric presentations are rare and may include anxiety, depressive syndrome, phobia, and nocturnal nightmare [27].

Prevalence of hypothermia was also elevated (39%) in our series, consistent with observations from French Polynesia [17,26] and Guadeloupe [7]. Interestingly, hyperthermia is considered in Polynesia as a symptom excluding the diagnosis of CP [9]. Moreover, ciguatera is named “disease of the dead” by populations from the Marquesas Islands, French Polynesia, in their traditional dialect, due to CP-induced body cooling. In addition, joint and muscle disorders such as arthromyalgia affecting the lower limbs are frequently observed, consistent with our series. Similarly, intense pruritus called “la grate” in the Pacific region (mainly in New Caledonia) is characteristic and may induce scratching lesions.

Like in our series, persistent manifestations occurring in ~20% of the patients may be neurological, cutaneous, and/or musculoarticular with almost constant asthenia [30,31]. By contrast, only recurrent neurological manifestations have been reported following the ingestion of non-ciguatoxic food such as poultry, pork, peanuts, and alcohol, or during physical activities including sexual activity [32].

Consistent with the literature [12,13], CP patient management was mainly supportive in our series. However, due to the absence of institutional guidelines, there was a great disparity in treatments. Atropine was used in poorly tolerated severe bradycardia of <60 bpm, as recommended [33]. Despite evidence of low quality to support its effectiveness to limit neuronal edema and impaired action potentials [34], mannitol was used in the early phase including itching and sensitive neurological syndrome at the request of the physician in charge. If needed, patients admitted to the emergency department were hospitalized shortly.

### 3.3. Fish Involved in CP

All kinds of fish in Martinique are likely to transmit ciguatera. Implicated finfish are mostly carnivorous such as trevallies, snappers, and barracudas. In French Polynesia, grazers are the major species, although carnivorous were also reported in the largest health authority series including 384 CP cases in 2018 [9]. Ciguatoxins enter the food chain through coral-eating fish and herbivores that graze the algae on which *Gambierdiscus* spp. and *Fukuoya* spp. are attached, as mainly observed in the coral reef in the Pacific Ocean, around the French Polynesia islands. Then, grazers are prey to omnivorous and carnivorous fish. Toxins accumulate along the food chain, so carnivores have higher ciguatotoxin levels than herbivores, and are thus more frequently responsible for CP, as in Martinique.

In our series, we found several CP cases related to unusual fish species. Mollusks such as lambi (*Strobus gigas*) were shown to transmit ciguatera in Polynesia (Benitiers, Sea Urchins, Trocas) [9,35]. Grazing marine snail-related CP was reported in Japan [36]. However, the absence of remains of the incriminated lambi prevented us from definitive analytical confirmation. Similarly, salmon (*Salmo salar*) was the source of CP, although no laboratory confirmation could be obtained. Interestingly, two grouped cases of CP were related to tuna from local fisheries, causing analytically confirmed six poisonings according to the statement of the local government. To date, this pelagic species is not considered as a ciguatera vector, the only other CP case linked to tuna ingestion reported in Guadeloupe in 2008 [25].

Ciguatoxins are non-protein fat-soluble toxins that accumulate in the head and viscera [37]. Interestingly, in the case of the edible parts of a grouper, *Variola louti*, the flesh ciguatoxin levels in the head were not higher than those in the filet [38]. However, tissue surrounding the eyeballs was more toxic than the flesh. In our series, trevallies seem to have caused more severe intoxication leading to a threefold increase in the risk of developing chronic symptoms (Table 4). Trevallies are cooked traditionally in broth or in blaff, which involves consumption of the head and viscera more often, whereas other fish are cooked by grilling or frying with consumption limited to the flesh.

### 3.4. Study Limitations

Our study has limitations related to its retrospective single-center design. Biases in the estimation of ciguatera incidence in Martinique may have resulted from the non-inclusion of patients with mild symptoms who did not consult a medical professional and the uncomplete recovery of hospital cases due to the absence of exact coding in the database. Since the most severe cases were included, the severity of CP may have been enhanced. Missing data were noted in some records, however; this was minimized by a systematic phone call to the patient when possible. Vital parameters at the time of intoxication were not available in RHA declaration forms. Follow-up was limited to 59% of the patients. In the absence of reliable routine bedside screening tests of ciguatoxin, diagnosis was based on history and clinical manifestations suggestive of CP. No analytical confirmation was obtained, which would have required analysis of the remains of meals in a specialized laboratory [39]. Except for the 19 grouped cases attributed to the consumption of snappers imported from the Indian Ocean area in 2017, which might have implicated I-CTX, comparisons according to the origin of the implicated fish (i.e., Caribbean sea *versus* the Indian Ocean) to identify significantly different clinical features in relation to the implicated toxins (i.e., C-CTXs *versus* I-CTX) were impossible, since the origin of the fish was not certain based on our retrospective data analysis. Similarly, misidentification of fish species could not be ruled out, although we believe that description and confirmation by physicians in charge and RHA specialists were reliable. No additional subgroup analysis could be performed especially regarding unusual cases of fish, due to the small case number and the retrospective study design with lack of some useful descriptive details.

Hence, encouraged by our findings, pieces of fish suspected to be ciguatera-contaminated are now sent systematically for laboratory analysis by the French Directorate of Food, Agriculture & Forestry, and the French Agency for Food, Environmental and Occupational Health & Safety. Implementation of guidelines for CP management should allow harmonizing the practices and offering optimized management. Our study could also support future preventive actions. We showed the great disparity in species causing CP in Martinique, including classic species such as snappers and jacks and rarer pelagic species such as tuna and sea bream. Grouped CP cases observed almost yearly suggest that fish caught in Martinique are potential vectors of ciguatera. The local population should be reminded not to eat the head or viscera of trevallies.

## 4. Conclusions

CP is an increasing, serious public health issue in Martinique, similar to other West Indies countries. The clinical picture is characterized by predominant gastrointestinal and neurological manifestations with possible life-threatening cardiovascular disturbances, although presentation may vary from one region to another in relation to the involved ciguatoxin type. The outcome is excellent despite possible manifestation persistence beyond 15 days.

## 5. Materials and Methods

### 5.1. Study Design and Data Collection

We performed a retrospective single-center observational study over a six-year period (1 October 2012 to 30 September 2018) including all CP patients managed by the emergency medical assistance service (SAMU 972) and/or admitted to the emergency department of the university hospital of Martinique as well as all CP patients declared to the RHA.

The study was conducted according to the Helsinki principles, declared and approved by our institutional review board. Patients were informed and invited to express their opposition to the use of their anonymized data if desired, but written consent was waived due to the retrospective and non-interventional methodology of the study.

CP was defined as a case of fish consumption followed within 48 h by gastrointestinal, neurological or cardiovascular manifestations [13]. Symptoms were defined as “persistent” if present after the acute phase that usually lasts 1–14 days after fish ingestion. For analysis, we considered trevallies (*Carangidae*) as a family of ray-finned fish, which includes jacks, pompanos, jack mackerels, runners, and scads.

Patients admitted to the hospital were identified using the electronic database of the hospital medical information department and SAMU 972 (i.e., PMSIpilot, Groupe PSIH, Lyon, France and Centaure™ software, Inetum, St-Ouen, France, respectively). For each patient, the age, gender, history, date of intoxication, incriminated fish species, number of intoxicated persons, clinical manifestations, treatments administered at the hospital, and follow-up data were collected. Of note, patients reported to the RHA received an individual questionnaire accompanied by a systematic phone call to collect all required information. CP patient follow-up at day 15 post-intoxication and outcome evaluation was performed during the study period by a consultation with a clinical toxicologist at the hospital or by phone call if required.

### 5.2. Statistical Analysis

Quantitative variables are expressed as median (interquartile range) and qualitative variables as numbers (percentages). Univariate comparisons between CP patients intoxicated by trevallies versus other fish were performed using Fisher’s exact tests for qualitative variables and Mann-Whitney U-tests for quantitative variables. ORs and their 95% CIs were calculated. Statistical analysis was performed using R statistical software version 3.6.3. A (Creative Commons Attribution-ShareAlike 4.0 International license, https://www.r-project.org/about.html, accessed on 4 April 2022) *p*-value < 0.05 was considered as statistically significant.

## Figures and Tables

**Table 1 toxins-14-00535-t001:** Clinical manifestations in 149 patients diagnosed with ciguatera poisoning in Martinique, French West Indies, in 2012–2018.

Clinical Characteristics	Prevalence or Value
**General manifestations**	**135/149 (91%)**
Body temperature ^†^	37.0 °C (36.0–37.1)
Myalgia	94/149 (63%)
Pruritus	94/149 (63%)
Asthenia	61/149 (41%)
Hypothermia (temperature ≤36.5 °C) *	26/66 (39%)
Arthralgia	54/149 (36%)
Chills	49/149 (33%)
Headaches	42/149 (28%)
Dizziness	34/149 (23%)
Malaise	28/149 (19%)
Sweating	6/149 (4%)
Dyspnea	5/149 (3%)
Limb edema	5/149 (3%)
**Gastrointestinal manifestations**	**134/149 (90%)**
Diarrhea	119/149 (80%
Abdominal pain	102/149 (69%)
Nausea	88/149 (59%)
Vomiting	80/149 (54%)
**Neurological manifestations**	**108/149 (72%)**
Paresthesia	77/149 (52%)
Dysgeusia	45/149 (30%)
Impairment of feeling of hot and cold	40/149 (27%)
Touch disorder	37/149 (25%)
Dysuria	25/149 (17%)
Reversal of hot and cold	24/149 (16%)
Balance/coordination/language impairment	24/149 (16%)
Visual disturbance	23/149 (15%)
Behavioral disorder	17/149 (11%)
Pain in cold	11/149 (7%)
**Cardiovascular manifestations**	**33/149 (22%)**
Heart rate ^†^	69 bpm (50–87)
Systolic blood pressure ^†^	112 mmHg (97–130)
Diastolic blood pressure ^†^	68 mmHg (59–78)
Bradycardia (heart rate <60 bpm) *	24/66 (36%)
Hypotension (systolic blood pressure <90 mmHg) *	10/66 (15%)
ECG abnormalities *	10/66 (15%)
Severe cardiovascular features ^#,^*	8/66 (12%)
Palpitations	5/149 (3%)
**Chronic manifestations ****	**31/77 (40%)**
Chronic pain	20/77 (26%)
Chronic neurological manifestations	15/77 (20%)
Chronic asthenia	11/77 (14%)
Chronic pruritus	10/77 (13%)

^†^ expressed as median (interquartile range); ^#^ severe cardiovascular features were defined as heart rate <40 bpm or systolic blood pressure <80 mmHg on admission; * determined in the 66 patients managed in the emergency department with available vital signs and electrocardiograms; ** determined in the 77 patients followed-up at the clinical toxicology consultation (*n* = 39) or by phone call (*n* = 38).

**Table 2 toxins-14-00535-t002:** Management in sixty-six patients diagnosed with ciguatera poisoning and managed in the emergency department of the university hospital of Martinique, French West Indies, in 2012–2018.

Management at the Emergency Department	Prevalence or Value
Supportive care	42/66 (64%)
Fluids (0.9% NaCl)	11/66 (17%)
20% mannitol infusion	11/66 (17%)
Intravenous atropine administration	10/66 (15%)
Hospitalization	24/149 (16%)
Length of hospital stay ^†^	1 days (1–2)
Follow-up at the clinical toxicology consultation	39/66 (59%)

^†^ expressed as median (interquartile range).

**Table 3 toxins-14-00535-t003:** Ingested fish that caused ciguatera poisoning in 149 patients in Martinique, French West Indies, in 2012–2018.

Incriminated Fish	Number of Incriminated Fish (N = 85)	Number of Intoxicated Patients (N = 149)
*Carangidae* (trevallies)	50	87
*Lutjanus buccanella* (snapper)	14	19
*Scomberomorus regalis* (king mackerel)	4	10
*Gymnothorax funebris* (moray eel)	3	4
*Thunnus atlanticus* (tuna)	2	6
*Coryphaena hippurus* (sea bream)	2	3
*Syphyraena barracuda* (barracuda)	1	9
*Epinephelus morio* (grouper)	1	1
*Strobus gigas* (spider conchs)	1	1
*Salmo salar* (salmon)	1	1
Mixed fish	1	4
Not described	4	4

**Table 4 toxins-14-00535-t004:** Comparison of clinical characteristics according to the fish responsible for ciguatera poisoning in 145 patients in Martinique ^†^, French West Indies, in 2012–2018.

Clinical Characteristics	Patients Intoxicated by Trevallies (*Carangidae*) (*n* = 87)	Patients Intoxicated by Other Fish Species (*n* = 58)	*p*-Value
Fish head or viscera ingestion (N = 144)	20/86 (24%)	5/58 (9%)	0.03
Acute vomiting (N = 145)	64/87 (74%)	23/58 (40%)	<0.0001
Acute nausea (N = 145)	61/87 (70%)	24/58 (41%)	0.001
Acute abdominal pain (N = 145)	68/87 (78%)	31/58 (53%)	0.002
Acute hypotension (N = 64)	14/35 (40%)	5/29 (17%)	0.05
Mannitol administration (N = 63)	8/33 (24%)	2/30 (6%)	0.09
Persistent symptoms (N = 74)	19/36 (50%)	9/36 (25%)	0.03

^†^ Four patients with unknown ingested fish were excluded from the analysis; persistent symptoms if present after the acute phase that usually lasts 1–14 days after fish ingestion.

## Data Availability

Not applicable.

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
