# Peer review of "Clinical Characteristics of Ciguatera Poisoning in Martinique, French West Indies—A Case Series"

_toxins, 2022, doi:10.3390/toxins14080535_

Round 1

Reviewer 1 Report

This manuscript described the clinical features of ciguatera poisonings in Martinique, French West Indies. The paper provides important information to figure out the CP in this region.

I found some lack of information and data, such as the number of cases, related patient information, and implicated fishes, including analytical data on the CTX levels. However, the accumulation of such data is important even in limited information.

I think this manuscript should be published in Toxins after being revised.

My comment and suggestions are as follows.

Major comments

  1. It has been well documented a wide range of symptoms and regional characteristics in CP. As one of the reasons, differences in types of ciguatoxins have been suspected, i.e., analogs of C-CTX from the Caribbean Sea, CTX1B and CTX3C from the Pacific, and I-CTX from the Indian Oceans. These toxins bind to the VGSC at the same site, however, they have different structures and could give different modes of action. Describe types of CTXs in the Introduction section and C-CTXs are present in the Caribbean Sea.
  2. Also, describe the characteristics of each region with references in Introduction section and rewrite the Discussion section to develop the discussion with this point of view.
  3. The FAO/WHO experts meeting recommended using "ciguatera poisoning" since ciguatera is not only caused by the finfish. The manuscript includes a case of poisoning by a marine snail. Replace "ciguatera fish poisoning" and "CFP" with "ciguatera poisoning" and "CP", respectively, throughout the manuscript.
  4. The cases included the due to the consumption of imported fish (L137-L138). These fishes were imported from the Indian Ocean, and the implicated toxins might be I-CTX. As I mentioned above, clinical features might differ from those caused by C-CTXs. I recommend classifying the cases into two groups by the origin of the implicated fish, the Caribbean and the Indian Ocean. The cases related to the Indian Ocean should be separated. Thus I recommend rewrite the sections of results and discussions.
  5. Implicated finfish were carnivorous, it should be mentioned. In French Polynesia, grazers are major species.

Respective comments

L34

“Gambier discus” should be as “Gambierdiscus”

L34-37

The primary substrate of Gambierdiscus is macroalga, not live corals. In the area where live corals are developed, the macroalgae are limited. Certain fast-growing algae increase after the live coral is destroyed for artificial or natural reasons. And Gambierdiscus may grow on the surface of the algae.

L39

“ciguatera” -> “ciguatoxin”?

L43

I think the number was suspected cases

L44

“its incidence sharply increased during the last decades in the West Indies.” Cite the references of sources.

L57

Spell out RHA

L57-58

“Three patients (2%) had a previous CFP history.”

Describe about these patients. Are their symptoms severer than other people shared with same dishes? Are their developed CP after consumed uncommon fish such as salmon? etc.

L60 and other parts

“three contaminated guests [3-5] per event” Since the style is same as citation and readers may confuse. Change the style to make sure what it means. Similar issue style are present other parts and they also should be changed.

L66
“Vital parameters available in 66/149 patients” Were these patients transferred to Emergency Department? If so, rewrite the sentence to mention it.

L78-L80

Describe what had been done for the rest of the patients. (no treatment?)

L82

“at day 15”

Dose it mean after onset or consumption?

L84

“consisted pain” describe what kind or which part?

L85

“pruritus” in whole body or particular parts?

L91

Should be removed

L94

“Rare cases were attributed to tunas (Thunnus atlanticus).”

Describe in detail. Were there any differences from typical cases? Collected area was recognized as danger area?

L95

“head or viscera”

Can you show respective number and percentage? CTX levels in the viscera may higher than the head.

Table 3

Regarding to unusual cases such as tuna, salmon, and conchs, can you describe in detail? Were they on markets?

Was there any possibility mis identification of fish species?  

The patients previously experienced ciguatera?

And so on

L110

Reference should be needed

L135-140

These portions should be described in Results section

L143

“In” -> “in”

L167

Insert “, French Polynesia” after “Marquesas Islands” And this sentence needed reference.

L170

"la grate" in the Pacific region

Mention which area of the Pacific.

L186-L188

Grazing marine snail related CP was reported from Japan in following manuscript.

Yoshiro HASHIMOTO, Shoji KONOSU, Masaki SHIBOTA, Katsuko WATANABE, Toxicity of a Turban-shell in the Pacific, NIPPON SUISAN GAKKAISHI, 1970, Volume 36, Issue 11, Pages 1163-1171

https://doi.org/10.2331/suisan.36.1163

L192

How it was confirmed?

N2a assay on the food ruminants?

L195

I wonder that we can say fatty for “head and viscera”

In the case of the flesh of grouper, Variola louti, the flesh of CTXs levels in the head were not higher than those in filet. However, tissue surrounding eye ball was more toxic than flesh.

Oshiro N, Nagasawa H, Kuniyoshi K, Kobayashi N, Sugita-Konishi Y, Asakura H, Yasumoto T. Characteristic Distribution of Ciguatoxins in the Edible Parts of a Grouper, Variola louti. Toxins. 2021; 13(3):218.

https://doi.org/10.3390/toxins13030218

Table 4

“Persistent symptoms”

Definition of “persistent” should be mentioned.

Figure S1

Show the location area in the Caribbean Sea and indicate the name of the district at least mentioned in the main text. I could not find figure legends.

Author Response

Reviewer 1-

This manuscript described the clinical features of ciguatera poisonings in Martinique, French West Indies. The paper provides important information to figure out the CP in this region.

I found some lack of information and data, such as the number of cases, related patient information, and implicated fishes, including analytical data on the CTX levels. However, the accumulation of such data is important even in limited information. I think this manuscript should be published in Toxins after being revised. My comment and suggestions are as follows.

=> We would like to thank the reviewer for his helpful comments to improve our manuscript.

 Major comments

  1. It has been well documented a wide range of symptoms and regional characteristics in CP. As one of the reasons, differences in types of ciguatoxins have been suspected, i.e., analogs of C-CTX from the Caribbean Sea, CTX1B and CTX3C from the Pacific, and I-CTX from the Indian Oceans. These toxins bind to the VGSC at the same site, however, they have different structures and could give different modes of action. Describe types of CTXs in the Introduction section and C-CTXs are present in the Caribbean Sea.

=> Done as suggested, we briefly added the description of the different known ciguatoxins and their relationships to the variability in CP expression, as follows: “A wide range of symptoms and regional characteristics have been attributed to CP. As one of the reasons, differences in types of ciguatoxins have been suspected [14]. At least sixteen types of ciguatoxins have been described according to their carbon skeleton and origin. The Pacific ciguatoxins (P-CTX) from the Pacific Ocean, classified in two types P-CTX I (also known as CTX4A type) and II (also known as CTX3C type), the Caribbean-ciguatoxin (C-CTX) found in Caribbean Sea and the Indian-ciguatoxin (I-CTX) in the Indian Ocean are the most prevalent types [15]. All these toxins bind to the site 5 of the α-subunit of the voltage gated sodium channels (VGSC) causing a negative shift in the voltage dependence of the channel activation and a concentration–dependent decrease in the maximum inward sodium current, which increases cell excitability. However, due to their variable structures, ciguatoxins could give different modes of action and result in variable clinical expression.” We also cited two additional references fore this paragraph.

  1. Also, describe the characteristics of each region with references in Introduction section and rewrite the Discussion section to develop the discussion with this point of view.

=> We briefly added sentences in the discussion section to report differences in clinical presentation that could be attributed to the variety of implicated toxins, as follows: “Variation in symptom patterns in the Pacific Ocean, Indian Ocean, and Caribbean Sea reflects geographic differences in ciguatoxin profiles and individual risk factors among patients in the different parts of the world [5]. In the Caribbean area, gastrointestinal manifestations are characteristic in the acute phase, followed by neurological, mainly peripheral, manifestations. By contrast, in the Pacific and Indian Ocean regions, neurological manifestations are more prevalent and severe in the acute phase.

  1. The FAO/WHO experts meeting recommended using "ciguatera poisoning" since ciguatera is not only caused by the finfish. The manuscript includes a case of poisoning by a marine snail. Replace "ciguatera fish poisoning" and "CFP" with "ciguatera poisoning" and "CP", respectively, throughout the manuscript.

=> Done as suggested.

  1. The cases included the due to the consumption of imported fish (L137-L138). These fishes were imported from the Indian Ocean, and the implicated toxins might be I-CTX. As I mentioned above, clinical features might differ from those caused by C-CTXs. I recommend classifying the cases into two groups by the origin of the implicated fish, the Caribbean and the Indian Ocean. The cases related to the Indian Ocean should be separated. Thus, I recommend rewrite the sections of results and discussions.

=> We agree with the reviewer that the origin of the fish could influence the clinical presentation. However, based on our retrospective method, it was almost impossible to be sure about the origin of the fish responsible for the CP even if bought from local markets, except for the 19 grouped cases in 2017. Therefore, we could not separate the patients according to the origin of the implicated fish (Caribbean versus Indian Ocean). This issue was addressed in the limitation section also as follows: Except for the 19 grouped cases attributed to the consumption of snappers imported from the Indian Ocean area in 2017, which might have implicated I-CTX, comparisons according to the origin of the implicated fish (i.e., Caribbean sea versus Indian Ocean) to identify significantly different clinical features in relation to the implicated toxins (i.e., C-CTXs versus I-CTX) were impossible, since the origin of the fish was not certain based on our retrospective data analysis.”

  1. Implicated finfish were carnivorous, it should be mentioned. In French Polynesia, grazers are major species.

=> Added to the discussion section as recommended.

 Respective comments

 L34 “Gambier discus” should be as “Gambierdiscus”

=> Changed as requested.

L34-37 The primary substrate of Gambierdiscus is macroalga, not live corals. In the area where live corals are developed, the macroalgae are limited. Certain fast-growing algae increase after the live coral is destroyed for artificial or natural reasons. And Gambierdiscus may grow on the surface of the algae.

=> Changed as requested, in addition to the other reviewers’ remarks to: “CP mostly results from accidental ingestion of fish flesh contaminated with the toxins produced by two dinoflagellate genus called Gambierdiscus spp. and Fukuoya [4]. These unicellular algae may grow on the surface of macroalgae, which proliferation is usually limited in the area where live corals are developed. However, if coral is destroyed for artificial or natural reasons, as more and more frequently observed with cyclones, sand fog, volcanic activity, sargassum waves, pollution, and work on the reefs [3-6], certain fast-growing algae may increase, stimulating the development of Gambierdiscus spp. and Fukuoya.

L39 “ciguatera” -> “ciguatoxin”

=> Changed as requested.

L43 - I think the number was suspected cases

=> Changed as requested.

L44 “its incidence sharply increased during the last decades in the West Indies.” Cite the references of sources.

=> Cited as requested.

L57 Spell out RHA

=> Done as requested.

L57-58 “Three patients (2%) had a previous CFP history.” Describe about these patients. Are their symptoms severer than other people shared with same dishes? Are their developed CP after consumed uncommon fish such as salmon? etc.

=> Developed as follows: “Of note, the two patients with previous CFP history did not exhibit more severe symptoms and did not consume uncommon fish.” 

L60 and other parts: “three contaminated guests [3-5] per event” Since the style is same as citation and readers may confuse. Change the style to make sure what it means. Similar issue style are present other parts and they also should be changed.

=> Changed everywhere as requested.

 L66: “Vital parameters available in 66/149 patients” Were these patients transferred to Emergency Department? If so, rewrite the sentence to mention it.

=> This was added to the manuscript for clarification. However, it was already mentioned in the second sentence of the result section.

L78-L80: Describe what had been done for the rest of the patients. (no treatment?)

=> Explained as follows: “In the other 83 not transported patients, no particular treatment was required or prescribed on the scene.”

L82: “at day 15” Dose it mean after onset or consumption?

=> The information was given in the manuscript.

L84 “consisted pain” describe what kind or which part?

L85 “pruritus” in whole body or particular parts?

=> Expanded as required in the result section.

L91 Should be removed; L94 “Rare cases were attributed to tunas (Thunnus atlanticus).”Describe in detail. Were there any differences from typical cases? Collected area was recognized as danger area?

=> It was impossible to differentiate cases related to tunas from the others due to their small number and the retrospective design of the study. No specific area of danger was recognized regarding tunas based on the available reports. 

L95 “head or viscera” Can you show respective number and percentage? CTX levels in the viscera may higher than the head.

=> We agree with the reviewer remark. However, it was impossible retrospectively to separate the two components.

 Table 3- Regarding to unusual cases such as tuna, salmon, and conchs, can you describe in detail? Were they on markets? Was there any possibility misidentification of fish species?  The patients previously experienced ciguatera? And so on

=> Due to the retrospective design of our study, it was impossible to give description that is more confident on cases related to unusual fish. Misidentification of fish species could not be ruled out, although we believe that description and confirmation by physicians and RHA specialists were reliable. To address the reviewer’s remarks, we added the following sentence in the limitation section: “Misidentification of fish species could not be ruled out although we believe that description and confirmation by physicians in charge and RHA specialists were reliable. No additional subgroup analysis could be performed especially regarding unusual cases of fish, due to the small case number and the retrospective study design with lack of some useful descriptive details.”

L110 Reference should be needed => Done as requested.

L135-140 These portions should be described in Results section => Done as requested.

L143“In” -> “in” => Corrected

L167 Insert “, French Polynesia” after “Marquesas Islands” And this sentence needed reference.

=> Done as requested. This expression come from the traditional dialect of local population. This information was added.

L170 "la grate" in the Pacific region. Mention which area of the Pacific.

=> Done as requested.

L186-L188 Grazing marine snail related CP was reported from Japan in following manuscript. Yoshiro HASHIMOTO, Shoji KONOSU, Masaki SHIBOTA, Katsuko WATANABE, Toxicity of a Turban-shell in the Pacific, NIPPON SUISAN GAKKAISHI, 1970, Volume 36, Issue 11, Pages 1163-1171 https://doi.org/10.2331/suisan.36.1163

=> The following reference was added.

 L192 - How it was confirmed? N2a assay on the food ruminants?

=> The official document from the regional health authorities referenced in the manuscript did not give additional details about the diagnostic procedure. Possibly N2a assay on the food ruminants. We therefore did not add uncertain information.

 L195 - I wonder that we can say fatty for “head and viscera”. In the case of the flesh of grouper, Variola louti, the flesh of CTXs levels in the head were not higher than those in filet. However, tissue surrounding eye ball was more toxic than flesh.

Oshiro N, Nagasawa H, Kuniyoshi K, Kobayashi N, Sugita-Konishi Y, Asakura H, Yasumoto T. Characteristic Distribution of Ciguatoxins in the Edible Parts of a Grouper, Variola louti. Toxins. 2021; 13(3):218.https://doi.org/10.3390/toxins13030218

=> As advised, we added the following reference and sentence in the manuscript: “Interestingly, in the case of the edible parts of a grouper, Variola louti, the flesh ciguatoxin levels in the head were not higher than those in the filet [34]. However, tissue surrounding the eyeballs was more toxic than the flesh.”

Table 4 -“Persistent symptoms”. Definition of “persistent” should be mentioned.

=> We added the definition present in the method section as footnote.

 Figure S1 - Show the location area in the Caribbean Sea and indicate the name of the district at least mentioned in the main text. I could not find figure legends.

=> Done as requested

Reviewer 2 Report

This article deals with ciguatera fish poisoning, and enhances our understanding on the risks, supportive-management and clinical outcome of this intoxication in Martinique, the largest French island in the West Indies. The article presents a retrospective, single-center observational study, obtained during the period 2012 to 2018, at the location indicated. Data, to some extent data are similar to that obtained in other Caribbean locations. Some points in the Abstract and Introduction need important revision as specified here below:

Line 6 abstract, it should be ciguatoxins, there are at least 16 ciguatoxins and not just one!!

Line 7, abstract, Note that ciguatoxins have also action on voltage-gated potassium channels.

Therefore, this should be mentioned in the abstract, since such an action affects sensory and motor neuronal systems, and is responsible for some of the CFP symptoms reported.

see

Mattei C, Marquais M, Schlumberger S, Molgó J, Vernoux JP, Lewis RJ, Benoit E. Analysis of Caribbean ciguatoxin-1 effects on frog myelinated axons and the neuromuscular junction. Toxicon. 2010 Oct;56(5):759-67. doi: 10.1016/j.toxicon.2009.07.026. Epub 2009 Jul 29. PMID: 19646468.

Schlumberger S, Mattei C, Molgó J, Benoit E. Dual action of a dinoflagellate-derived precursor of Pacific ciguatoxins (P-CTX-4B) on voltage-dependent K(+) and Na(+) channels of single myelinated axons. Toxicon. 2010 Oct;56(5):768-75. doi: 10.1016/j.toxicon.2009.06.035. Epub 2009 Jul 7. PMID: 19589350.

Hidalgo J, Liberona JL, Molgó J, Jaimovich E. Pacific ciguatoxin-1b effect over Na+ and K+ currents, inositol 1,4,5-triphosphate content and intracellular Ca2+ signals in cultured rat myotubes. Br J Pharmacol. 2002 Dec;137(7):1055-62. doi: 10.1038/sj.bjp.0704980. PMID: 12429578; PMCID: PMC1573594.

Line 34, should be written "Gambierdiscus" in a single word, note that not only the genus Gambierdicus has been reported, but also Fukuoya. This should be mentioned and pertinent references should be quoted.

Line 46, Note that ciguatoxins in addition to act on voltage-gated sodium channels also act on voltage-gated potassium channels. Please add this information and quote references.

Author Response

This article deals with ciguatera fish poisoning, and enhances our understanding on the risks, supportive-management and clinical outcome of this intoxication in Martinique, the largest French island in the West Indies. The article presents a retrospective, single-center observational study, obtained during the period 2012 to 2018, at the location indicated. Data, to some extent data are similar to that obtained in other Caribbean locations. Some points in the Abstract and Introduction need important revision as specified here below:

=> We would like to thank the reviewer for his helpful remarks.

Line 6 abstract, it should be ciguatoxins, there are at least 16 ciguatoxins and not just one!!

=> Corrected.

Line 7, abstract, Note that ciguatoxins have also action on voltage-gated potassium channels. Therefore, this should be mentioned in the abstract, since such an action affects sensory and motor neuronal systems, and is responsible for some of the CFP symptoms reported.

see

Mattei C, Marquais M, Schlumberger S, Molgó J, Vernoux JP, Lewis RJ, Benoit E. Analysis of Caribbean ciguatoxin-1 effects on frog myelinated axons and the neuromuscular junction. Toxicon. 2010 Oct;56(5):759-67. doi: 10.1016/j.toxicon.2009.07.026. Epub 2009 Jul 29. PMID: 19646468.

Schlumberger S, Mattei C, Molgó J, Benoit E. Dual action of a dinoflagellate-derived precursor of Pacific ciguatoxins (P-CTX-4B) on voltage-dependent K(+) and Na(+) channels of single myelinated axons. Toxicon. 2010 Oct;56(5):768-75. doi: 10.1016/j.toxicon.2009.06.035. Epub 2009 Jul 7. PMID: 19589350.

Hidalgo J, Liberona JL, Molgó J, Jaimovich E. Pacific ciguatoxin-1b effect over Na+ and K+ currents, inositol 1,4,5-triphosphate content and intracellular Ca2+ signals in cultured rat myotubes. Br J Pharmacol. 2002 Dec;137(7):1055-62. doi: 10.1038/sj.bjp.0704980. PMID: 12429578; PMCID: PMC1573594.

=> Added as requested.

Line 34, should be written "Gambierdiscus" in a single word, note that not only the genus Gambierdicus has been reported, but also Fukuoya. This should be mentioned and pertinent references should be quoted.

=> Modified as requested with an additional reference.

Line 46, Note that ciguatoxins in addition to act on voltage-gated sodium channels also act on voltage-gated potassium channels. Please add this information and quote references.

=> Added.

Reviewer 3 Report

The manuscript provides a descripyion of cases of ciguatera fish poisoning that are scarcelly found in the literature, thus, it is important to know, However some minor points should be addressed:

Add abbrevuations

Page 34, introduction Gambier discuss spp, should be Gambierdiscuss... Moreover, ciguatoxins are also produced by Fukuyoa especies, and this should be added.

The authors could disccus what is the risk of  blood lactate increased

Table 1: Prëvalence... change to prevalence

What is the reason to evaluate the body temperature in only 66 patients?

If one of the patognomic signs of ciguatera is the reversal of cold and hot sensations, and the authors only found this in 16 % of the patients, this fact should be addressed in the discussion section.

Please revise the references for formating, example, reference 1 states Eds 2022

Author Response

This manuscript describes the clinical characteristics of ciguatera fish poisoning in Martinique, French West Indies, during six years (2012-2018) and is of great interest to the readers of Toxins. The manuscript provides a description of cases of ciguatera fish poisoning that are scarcelly found in the literature, thus, it is important to know. However some minor points should be addressed.

=> We would like to thank the reviewer for his helpful remarks.

Add abbreviations

=> All abbreviations were recalled when first used. There is no possibility to add such a section according to the journal style.

Page 34, introduction Gambier discuss spp, should be Gambierdiscuss... Moreover, ciguatoxins are also produced by Fukuyoa especies, and this should be added.

=> Added and referenced.

The authors could disccus what is the risk of blood lactate increased

=> We briefly added the most probable interpretation of the elevation in lactate, as follows: “ …, suggestive of tissue hypoxia in relation to cardiovascular failure.”

Table 1: Prëvalence... change to prevalence

=> Changed

What is the reason to evaluate the body temperature in only 66 patients?

=> This information was only available in 66 patients admitted to the hospital. The records did not provide the information when the patients was examined at home or if health authorities declared the case.

If one of the patognomic signs of ciguatera is the reversal of cold and hot sensations, and the authors only found this in 16 % of the patients, this fact should be addressed in the discussion section.

=> Done as follows: “Interestingly, whereas sometimes acknowledged as pathognomonic of CP [27], reversal of hot and cold was only observed in 16% of our patients, underlying the rarity of specific clinical symptoms or signs that may unambiguously ascertain clinical diagnosis. Studies suggested that unusual sensations represent tingling or “electric shock” pain rather than a true reversal of hot and cold perception [29].”

Please revise the references for formating, example, reference 1 states Eds 2022

=> Corrected.

Reviewer 4 Report

This manuscript describes the clinical characteristics of ciguatera fish poisoning in Martinique, French West Indies, during six years (2012-2018) and is of great interest to the readers of Toxins. 

Author Response

=> We would like to thank the reviewer for his encouraging analysis.

Round 2

Reviewer 1 Report

The revised manuscript was improved.

First of all, I strongly recommend that show the line number(s) (e.g., L221-222) where revisions were made. It was very difficult and made me confused to find revised parts in the manuscript.

Comment on the responses to my previous comments:

L135-140 These portions should be described in Results section => Done as requested.
-> I could not find the revised portion.

L192 - How it was confirmed? N2a assay on the food ruminants?
=> The official document from the regional health authorities referenced in the manuscript did not give additional details about the diagnostic procedure. Possibly N2a assay on the food ruminants. We therefore did not add uncertain information.
->
At least, the authors mention who confirmed scientifically. E.g., "according to the statement of the local government, ...... were analytically confirmed"

Figure S1
I still do not understand where Martinique locate. Add a map showing the location of the island in Central America

Respective comments

L36 and L41
The genus name should be in Italic.
Confirm that it should be inserted "spp." after "Fukuyoa"

L44
I think "grazer" is not "large predetary".

L49
Is Ref 3 the correct citation?  This manuscript described dinoflagellate not the epidemiological aspect of CP.

L55
Confirm the ref. 14 is appropriate to cite or not.

L56
What does "sixteen types" mean? classes or compound?
more than 20 CTXs analogs were reported from the Pacific.
As the authors mentioned, 4 types CTXs were recognized.

L58
In ref 15, no classification "P-CTX I" nor "P-CTX II" was mentioned. It will make readers confused. re-write them.

L60-65
Need citation

L226-227
Need citation for "In French Polynesia ....".
I mentioned it to show an example, to show the differences between the study site and French Polynesia. If you refer to this, develop a discussion.

Author Response

The revised manuscript was improved. First of all, I strongly recommend that show the line number(s) (e.g., L221-222) where revisions were made. It was very difficult and made me confused to find revised parts in the manuscript.

R- We would like to thank the reviewer for all efforts provided to help us improving our manuscript.

Comment on the responses to my previous comments:

L135-140 These portions should be described in Results section => Done as requested. I could not find the revised portion.

R- We performed the suggested revision (highlighted in green, L88-91) and adapted the following sentence.

L192 - How it was confirmed? N2a assay on the food ruminants? => The official document from the regional health authorities referenced in the manuscript did not give additional details about the diagnostic procedure. Possibly N2a assay on the food ruminants. We therefore did not add uncertain information. At least, the authors mention who confirmed scientifically. E.g., "according to the statement of the local government, ...... were analytically confirmed"

R- We rephrased our sentence as recommended by the reviewer (L258-259).

Figure S1
I still do not understand where Martinique locate. Add a map showing the location of the island in Central America

R- We add a map to clarify the location of Martinique in Central America (L367). 

Respective comments:

L36 and L41- The genus name should be in Italic. Confirm that it should be inserted "spp." after "Fukuyoa"

R- Corrected (L54 and L59).

L44- I think "grazer" is not "large predetary".

R- We deleted the word “large” to avoid any confusion (L61).

L49- Is Ref 3 the correct citation?  This manuscript described dinoflagellate not the epidemiological aspect of CP.

R- We agree with the reviewer that our reference was not the one focusing on epidemiology. We changed as requested. The reference 5 was the adequate one (L66).

L55- Confirm the ref. 14 is appropriate to cite or not.

R- We felt that this basic science reference is useful to allow the reader understand the differences between the toxins regarding their electrophysiological effects (L72).

L56- What does "sixteen types" mean? classes or compound? more than 20 CTXs analogs were reported from the Pacific. As the authors mentioned, 4 types CTXs were recognized.

R- We corrected to “4 types” to avoid any misunderstanding (L72-73).

L58- In ref 15, no classification "P-CTX I" nor "P-CTX II" was mentioned. It will make readers confused. rewrite them.

R- To avoid any misunderstanding, we deleted the subdivision between "P-CTX I" nor "P-CTX II", as not useful to understand the manuscript (L74-76).

L60-65- Need citation

R- Done. We referred to the electrophysiological study (L76).

L226-227- Need citation for "In French Polynesia ....". I mentioned it to show an example, to show the differences between the study site and French Polynesia. If you refer to this, develop a discussion.

R- We add the reference on French Polynesia cases (i.e. [9] Chinain, et al.). We developed a short discussion as requested: “All kinds of fish in Martinique are likely to transmit ciguatera. Implicated finfish are mostly carnivorous such as trevallies, snappers, and barracudas. In French Polynesia, grazers are the major species, although carnivorous were also reported in the largest health authority series including 384 CP cases in 2018 [9]. Ciguatoxins enter the food chain through coral-eating fish and herbivores that graze the algae on which Gambierdiscus spp. and Fukuoya spp. are attached, as mainly observed in the coral reef in the Pacific Ocean, around the French Polynesia islands. Then, grazers are preys to omnivorous and carnivorous fish. Toxins accumulate along the food chain, so carnivores have higher ciguatotoxin levels than herbivores, and are thus more frequently responsible for CP like in Martinique.” (L242-250)

Academic Editor Comments:

The authors should add (in section 5.2) some information on the specific statistical package used for the statistical processing of their results. This can be done during the revision phase

R- Done as follows: “Statistical analysis was performed using R statistical software version 3.6.3.” (L344).